# Exosomal Plasma Gelsolin Is an Immunosuppressive Mediator in the Ovarian Tumor Microenvironment and a Determinant of Chemoresistance

**DOI:** 10.3390/cells11203305

**Published:** 2022-10-20

**Authors:** Toshimichi Onuma, Meshach Asare-Werehene, Yoshio Yoshida, Benjamin K. Tsang

**Affiliations:** 1Department of Obstetrics & Gynecology, Faculty of Medicine & Interdisciplinary School of Health Sciences, Faculty of Health Sciences, University of Ottawa, Ottawa, ON K1H 8L1, Canada; 2Department of Cellular and Molecular Medicine & the Centre for Infection, Immunity and Inflammation (CI3), Faculty of Medicine, University of Ottawa, Ottawa, ON K1H 8M5, Canada; 3Chronic Disease Program, Ottawa Hospital Research Institute, Ottawa, ON K1H 8L6, Canada; 4Department of Obstetrics and Gynecology, University of Fukui, Fukui 910-8507, Japan

**Keywords:** plasma gelsolin (pGSN), extracellular vesicles (EVs), chemoresistance, tumor microenvironment (TME), ovarian cancer, immune cells, T cells, macrophages, apoptosis

## Abstract

Ovarian Cancer (OVCA) is the most fatal gynecologic cancer and has a 5-year survival rate less than 45%. This is mainly due to late diagnosis and drug resistance. Overexpression of plasma gelsolin (pGSN) is key contributing factor to OVCA chemoresistance and immunosuppression. Gelsolin (GSN) is a multifunctional protein that regulates the activity of actin filaments by cleavage, capping, and nucleation. Generally, it plays an important role in cytoskeletal remodeling. GSN has three isoforms: cytosolic GSN, plasma GSN (pGSN), and gelsolin-3. Exosomes containing pGSN are released and contribute to the progression of OVCA. This review describes how pGSN overexpression inhibits chemotherapy-induced apoptosis and triggers positive feedback loops of pGSN expression. It also describes the mechanisms by which exosomal pGSN promotes apoptosis and dysfunction in tumor-killing immune cells. A discussion on the potential of pGSN as a prognostic, diagnostic, and therapeutic marker is also presented herein.

## 1. Introduction

### 1.1. Ovarian Cancer (OVCA) and Extracellular Vesicles (EVs)

OVCA is associated with the highest mortality rate among gynecological cancers; hence, the development of effective treatments for OVCA is critical. Approximately, 90% of OVCAs are epithelial with the most common subtypes being high-grade serous (HGS;70%), endometrioid (10%), clear cell (10%), mucinous (3%), and low-grade serous carcinomas (<5%). These pathologic subtypes differ in terms of epidemiological and genetic risks, precursor lesions, molecular events during cancer initiation, chemotherapy responses, and prognosis [1]. OVCA accounts for 3.4% of all cancer cases and 4.7% of all cancer deaths. Furthermore, OVCA is associated with lower incidence and death rates than those seen with corpus and cervical cancers [2]. However, among all gynecological cancers, OVCA has the highest mortality rate. The median age at diagnosis and 5-year survival rates associated with cervical, corpus, and OVCAs are 50 years and 66.7% [3], 63 years and 81.3% [4], and 63 years and 49.7% [5], respectively. Moreover, compared to cervical and corpus cancers, OVCA is associated with the worst prognosis. Early detection of OVCA is essential for successful treatment; however, current biomarkers are less reliable and inefficient in diagnosing patients at an early stage. Patients with OVCA show symptoms such as bloating, nausea, abdominal distention, changes in bowel function, urinary symptoms, back pain, fatigue, and loss of weight. Symptoms of OVCA occur 20–30 times per month, and they are considerably more severe with recent onset as compared to those with benign tumors and women visiting primary care clinics [6]. These symptoms are, however, nonspecific and can be unrelated to OVCA. Furthermore, there are no effective methods for detecting OVCA in the general population [7]. Consequently, 75% of patients with OVCA are diagnosed at stage III or stage IV where treatment is unlikely to be curative [8]. The probability of long-term survival (more than 10 years) for patients at stage I or II of OVCA is 80–95%, but only 10–30% for patients with stage III or IV disease [8].

Chemotherapy resistance is the main challenge in treating OVCA. Typically, patients with advanced stage OVCA are treated with primary debulking surgery and platinum-based chemotherapy [9]. Most OVCAs are initially responsive to platinum-based chemotherapies; however, 80% of initial responders acquire resistance [10]. This resistance is mediated by increased extrusion or decreased influx of drugs into the cell. Furthermore, resistance is also acquired by activating DNA repair mechanisms and detoxification proteins (cytochrome P450 complex) [11,12]. Chemotherapy gradually enables cancer cells to alter their genes, eventually acquiring resistance [11]. Moreover, several other mechanisms have been proposed for chemotherapy resistance in OVCA, related to cancer stem cells [13], autophagy [14], hypoxia, and endoplasmic reticulum stress [15]. A remarkable association has also been found between the extracellular vesicles (EVs) and chemotherapy resistance in malignant tumors [16,17]. EVs alter the immune system in the tumor microenvironment (TME), causing chemotherapy resistance [18].

It has been established that exosomes play a major role in chemoresistance and immune dysfunction in OVCA. By size and biosynthesis mechanism, EVs can be classified into different categories-exosomes (30–150 nm diameter derived from endosomes), microvesicles (150–1000 nm diameter derived from plasma membrane budding), apoptotic vesicles (1–5 µm diameter derived from apoptosis), and oncosomes (1–10 µm diameter derived from shedding of membrane blebs) [19,20,21]. One of the mechanisms of exosome release is the “classic pathway”, in which intraluminal vesicles (ILVs) are formed within the multivesicular endosomes. Their membranes, in turn, fuse with either the lysosome, which degrades cargo, or with the plasma membrane, releasing the ILVs called exosomes. Another mechanism of release includes direct budding of the plasma membrane, and is called as the “direct pathway” [21,22].

Exosomes play a key role in chemoresistance. Chemoresistant cells release cisplatin through exosomes [23,24]. OVCA cells produce EVs that promote both platinum resistance and OVCA cell invasiveness [25]. Most cancer cells secrete exosomes that are engulfed by proximal and distal recipient cells to convey internal signals [26]. A cargo of exosomes comprises various intracellular proteins, RNA, DNA, amino acids, lipids, and metabolic products [27]. Several miRNAs have also been implicated in drug resistance in OVCA, including miR-433 [28], miR-21-3p [29], miR-1246 [30], miR-98-5p [31], miR-223 [32], miR-21-3p [33], miR-891-5p [33], miR-21-5p [33,34], and miR-429 [35]. Exosomes secreted by OVCA cells in ascites fluid, cultured in hypoxic conditions, contain Signal transducer and activator of transcription 3 (STAT3) and FS7-associated cell surface antigen (FAS), which considerably increase cell migration and invasion and chemoresistance in vitro and tumor progression and metastasis in vivo [36]. Exosomes contain the transcription factor GATA Binding Protein 3 (GATA3) that influence macrophage interaction with HGS OVCA cells. These factors support the proliferation and motility, and confer cisplatin resistance in HGS OVCA cell lines with mutant TP53 [37]. This review focuses on plasma gelsolin (pGSN), which has recently gained attention as a mediator and marker of chemoresistance, immunoresistance, early diagnosis and a potential therapeutic target. Specifically, pGSN has been implicated in the following processes: (1) exosomal secretion and conferring of chemoresistance in otherwise chemosensitive OVCA cells; (2) autocrine and paracrine feedback mechanism of pGSN expression; and (3) exosomal-mediated downregulation of tumor-killing immune cells such as CD8+ T cells, CD4- T cells, M1 macrophages and natural killer cells.

### 1.2. Structure, Function, and Regulation of Gelsolin (GSN)

GSN is a multifunctional protein, primarily responsible for remodeling cytoskeletal structures. It is involved in actin filament severing, capping, and nucleating. These features of GSN have an important impact on cell shape, chemotaxis, and secretion [38,39]. GSN is encoded on chromosome 9 in humans, and its molecular weight ranges from 82–85 kDa. It has six homologous domains (G1-G6) that are responsible for its biological functions [40]. Several factors regulate GSN activity, including calcium [41], pH [42], and phosphatidylinositol 4,5-bisphosphate [43]. Three forms of GSN have been identified: cytosolic GSN (cGSN), pGSN, and gelsolin-3 although the first two are the most characterized [41,44] (Figure 1A). The previous study has identified that gelsolin-3 is mainly localized to oligodendrocytes in the brain. However, the function of gelsolin-3 remains unclear [44,45]. The N-terminus of pGSN contains an additional 24-amino acid residue which is absent in the other isoforms [41] (Figure 1B). Disulfide bonds between Cys188 and Cys201 in the G2 domain enhance the stability of the pGSN in extracellular environments [46,47,48]. Caspase 3 cleaves GSN at the linker, separating it into C-terminal GSN and N-terminal GSN [40] (Figure 1C). Human plasma concentrations of pGSN range between 100 and 330 ng/mL; however, these values may differ based on the detection method used. pGSN levels in serum are 24% lower than those in plasma because of interactions with fibronectin, fibrin, and fibronectin-like proteins [49]. cGSN is universally expressed, but cells expressing pGSN are less common [38]. Skeletal, cardiac, and smooth muscles are identified as the main sources of GSN in blood [50]; however, malignant tumor cells also produce higher levels of pGSN [51,52]. Detection of only cGSN expression in tumor tissues is challenging given the antibodies and strategies used target the c-terminal regions which is similar in both pGSN and cGSN. Thus, could be classified as total GSN rather than cGSN. Recent studies have specifically targeted pGSN with antibodies that binds to epitopes at the extra 24 amino acid sequence on the N-terminal of GSN which is absent in the cGSN [18,53].

## 2. pGSN and Chemoresistance

### 2.1. Autocrine and Paracrine Mechanisms of pGSN-Mediated Chemoresistance

Recent studies have demonstrated pGSN overexpression as a key factor in OVCA chemoresistance as well as other malignancies [18,51,52,53,54]. Most importantly, the exosomal secretion of pGSN has been observed to have pleotropic functions in the tumor microenvironment; responses that support tumor growth and suppress the immune system. Chemoresistant OVCA cells secrete higher levels of exosomal pGSN. Using electron microscopy, ELISA and Western blot, pGSN has been detected in exosomes derived from both chemosensitive and chemoresistant OVCA cells with more pGSN discovered in the chemoresistant-derived exosomes [52]. In a co-culture system, chemoresistant-derived exosomes (A2780cp, OV90 and OV866 (2)) conferred cisplatin resistance on otherwise chemosensitive OVCA cells (A2780s, OV4453 and OV2295) regardless of the histological subtype differences; a response that was abrogated when pGSN was silenced in the chemoresistant-derived exosomes. In contrast, chemosensitive-derived exosomes failed to induce cisplatin resistance in chemosensitive cells [52]. Pre-treatment of chemosensitive OVCA cells with exogenous human recombinant pGSN induced a resistant phenotype to cisplatin treatment. These findings suggest that pGSN mediates chemoresistance in OVCA cells via autocrine and paracrine mechanisms (Figure 2). Other signaling processes have been reported to be involved in exosome secretion. For example, reduced O-GlcNAcylation of Synaptosomal-associated protein 23 (SNAP-23) in SKOV3 cells leads to the formation of soluble N-ethylmaleimide-sensitive factor attachment protein receptors which further increases cisplatin efflux from the cell and stimulates exosome release, leading to chemoresistance [24]. Exosomal pGSN induces chemoresistance via an α5β1 integrins/FAK/Akt/HIF-α axis. This notion is supported by the observations that chemoresistant-derived exosomal pGSN increased HIF-1α expression in chemosensitive cells via Akt activation and that blocking the α5β1 integrin receptor attenuated the activation of the c-Met/Src/FAK signaling pathway in OVCA cells. This resulted in the downregulation of HIF1-α [52,55]. Additionally, pGSN-mediated cisplatin resistance in otherwise chemosensitive OVCA cells was abrogated by mutant FAK, suggesting that FAK plays a key role in OVCA chemoresistance [52]. Taken together, these findings suggest that exosomal pGSN enhances chemoresistance via α5β1 integrins/FAK/Akt/HIF-α axis in chemosensitive OVCA cells (Figure 2).

Interestingly, pGSN has been shown to upregulate its own gene expression endogenously via autocrine and paracrine means thus, making it a strong promoter of tumor growth and chemoresistance development. This unique feature of pGSN has recently been reported in OVCA. Upon exogenous treatment of OVCA cells with pGSN, HIF1α binding to pGSN promoter region is enhanced resulting in increased pGSN expression, exosomal packaging and exosomal secretion of pGSN (Figure 2); changes that lead to cisplatin resistance. The same findings were observed when chemosensitive OVCA cell were co-cultured with chemoresistant cells. Other feedback mechanisms have been reported in OVCA where DSCR8 overexpression promotes proliferation, invasion, and EMT and suppresses apoptosis in A2780 and SKOV3 cells. Additionally, a positive feedback loop is formed via the LncRNA DSCR8/miR-98-5p/STAT3/HIF-α axis [56].

### 2.2. Apoptosis Regulation by pGSN

Chemotherapeutic agent-induced apoptosis is an important indicator of chemoresponsiveness of cancer cells. There are pro-apoptotic and anti-apoptotic factors that regulate the ultimate responses of cancer cells to chemotherapeutic agents leading to improved patient prognosis and overall survival. A variety of mechanisms is involved in pGSN-mediated apoptosis which is also dependent on the surrounding conditions and cell types [40]. GSN has both pro-apoptotic and anti-apoptotic properties that contribute to cellular processes. In hepatocellular carcinoma cell line (SMMC7721), GSN overexpression suppressed apoptosis through the regulation of caspase-3 and Bcl-2 [57]. GSN overexpression also inhibits the release of cytochrome c from mitochondria and prevents the activation of caspase-3, -8, and -9, resulting in apoptosis inhibition [58]. Other studies have also reported the pro-apoptotic properties of GSN, in which smooth muscle cells isolated from GSN-deficient mice were found resistant to inflammatory cytokine-induced apoptosis [59]. Similarly, GSN overexpression in squamous cell carcinoma cell line, Tca8113 and TNF-resistant MCF-7 breast cancer cells have been shown sensitize their responsiveness to treatment [60,61].

In melanoma A7 cells, GSN is cleaved at the linker site by caspase-3 to produce two fragments; a 39-kDa N-terminal GSN (pro-apoptotic) and a 41-kDa C-terminal GSN (anti-apoptotic) [62,63]. The pro-apoptotic N-terminal of GSN causes structural changes and apoptosis in the melanoma A7 cells [62]. Similarly, native-Page experiments have shown that the N-terminal GSN inhibited DNase1-actin interaction resulting in apoptosis [64].

Chemotherapy resistance depends largely on the ability of cancer cells to resist apoptosis [65]. Fas-associated death domain-like interleukin-1b-converting enzyme-like inhibitory protein (FLIP) and Itchy E3 Ubiquitin Protein Ligase (ITCH) play critical roles in chemotherapy-induced apoptosis in OVCA cells [66,67]. FLIP is a major anti-apoptotic protein frequently overexpressed in solid tumors. FLIP is expressed as long (FLIP-L) and short (FLIP-S) splice forms and binds to the FAS-associated death domain protein that links FLIP to caspase-8 [68]. FLIP interacts with an E3 ligase (i.e., ITCH) in response to cis-Diamminedichloroplatinum (CDDP), resulting in ubiquitination and proteasomal degradation. This leads to apoptosis through the activation of caspase-8 and caspase-3 [66,67]. In OVCA, GSN induces cisplatin resistance by regulating the FLIP/ICTH/caspase-8/caspase-3 axis. FLIP expression is decreased in chemosensitive OV2008 cells but overly expressed in C13* cell lines, which is a chemo-resistant variant of OV2008 [69]. It has been hypothesized that, in chemosensitive condition, GSN forms a complex with FLIP-ITCH which is dissociated upon CDDP treatment. As a result, GSN is cleaved by activated caspase 3 leading to the production of a C-terminal fragment. Contrary to other studies, the C-terminal fragments of GSN produced in OVCA possesses pro-apoptotic properties hence sensitize chemoresistant cells to cisplatin-induced death. In chemoresistant conditions, the higher levels of GSN prevents CDDP from dissociating GSN from the FLIP-ITCH complex. This prevents caspase-3 and -8 activation and caspase-mediated GSN cleavage; responses that inhibit apoptosis in chemoresistant OVCA cells [70] (Figure 3).

## 3. pGSN and Immune Dysfunction

### 3.1. T Cell Dysfunction and Increased Glutathione (GSH) Production in OVCA through Decreased Interferon (IFN)γ Production

T cell infiltration into the tumor has a great impact on cancer prognosis, treatment responses and overall patient survival [18,71,72,73]. This also has a significant impact on immunotherapy response. In breast cancer, high tumor infiltration of CD8+ T cells is associated with complete pathologic response after chemotherapy [73]. Similarly, the ratio of tumor-infiltrating FOXP3 (+)/CD8(+) T cells is associated with responsiveness to chemotherapy, suggesting an important role of T cells in tumor elimination [71]. In OVCA, it has also been demonstrated that T cell infiltration results in improved progression-free survival (PFS) and overall survival (OS); responses that are mostly observed in chemosensitive patients compared to chemoresistant patients [18,72].

Recently, studies in OVCA and other cancer types have reported that EVs have immunosuppressive effects on immune cells such as T cells [74,75,76,77,78]. Chemoresistant OVCA cells (OV90 and A2780cp) cells produce increased levels of exosomal pGSN (compared with their sensitive counterparts) that suppresses interferon gamma (IFNγ) production as well as induces caspase-3-dependent apoptosis in CD8+ T cells [18]. Although CD8+ T cells were killed by exosomal pGSN, naïve CD4 T cells were preferentially polarized into type 2 helper T cells (Th2); responses that contribute to tumor growth and chemoresistance [18]. A shift in immune response from Th1 to Th2 in OVCA tissue, ascites fluid, and blood has been associated with a poor prognosis for patients [79,80]. Higher pGSN levels in blood is indicative of the amount of CD8+ T cells infiltrating tumor tissue in patients with OVCA [18]. These findings are consistent with a pancreatic cancer investigation in which pGSN secretion selectively killed CD8+ T cells but not CD4+ T cells [51] (Figure 4). The authors also reported that pGSN binds to CD37 and inactivates CD4+ T cells in prostate cancer whereas major histocompatibility complex (MHC) I-dependent cell–cell interaction upon pGSN sequestering results in the induction of apoptosis in activated CD8+ T lymphocytes [51].

The oxidative status of cancer cells is critical to their responsiveness to chemotherapy. Higher levels of antioxidants reduce the oxidative state of cancer cells thus, making them highly resistant to chemotherapeutic agents. Glutathione (GSH) is the most abundant non-protein thiol in all mammalian tissues and protects against oxidative stress. It is an important component of redox signaling, essential for detoxification of toxic substances and involved in cell proliferation, apoptosis, immunity, and fibrogenesis [81,82]. Given high levels of intracellular GSH chelate anticancer drugs, a reduced intracellular GSH levels are required to kill cancer cells [82]. pGSN increases intracellular GSH expression and causes chemotherapy resistance. GSH levels in blood were increased in irradiated mice after administration of recombinant human pGSN and improved their antioxidant status [83]. In chemoresistant OVCA cells, pGSN induces the phosphorylation of nuclear factor-erythroid factor 2-related factor 2 (NRF2) resulting in enhanced production of intracellular GSH. Increased GSH chelates and detoxifies CDDP as well as reduces intracellular CDDP accumulation (reduced γH2AX). These responses inhibit CDDP-mediated apoptosis in chemoresistant OVCA cells [18]. Other investigations have demonstrated that GSH released by fibroblasts reduces the accumulation of platinum in OVCA cells, making them more resistant to chemotherapy. By altering GSH and cysteine metabolism in fibroblasts, CD8(+) T cells attenuate this resistance mechanism.

In chemosensitive condition, there is an increased production of IFNγ, leading to the activation of the IFNGR1/JAK/STAT1 pathway; responses that suppress the GSH production and sensitize cells to CDDP-induced cell death [18]. Similarly, CD8+ T cell-derived IFNγ regulates fibroblast GSH and cysteine metabolism through upregulation of gamma-glutamyltransferases and transcriptional repression of system x_c_^−^ cystine and glutamate antiporter via the JAK/STAT1 pathway [84] (Figure 4).

### 3.2. pGSN and Macrophage Dysfunction

Tumor-associated macrophages (TAMs) are generally cells with an M2 phenotype that exert anti-inflammatory and tumor-promoting effects. M2 phenotype affects multiple steps of tumorigenesis, including tumor cell survival, proliferation, stemness, invasion, and angiogenesis. In addition, M2 macrophage cells inactivate tumor suppressor cells such as cytotoxic T cells and natural killer cells [85,86]. TAMs with the M1 phenotype are inflammatory and possess phagocytic properties. M1 macrophages can kill tumor cells by releasing nitric oxide (NO) and antibody dependent cellular cytotoxicity (ADCC) [86,87]. Exosomes can cause M1 macrophages dysfunction and promote M1 to M2 polarization in OVCA [88,89,90]. Exosomal pGSN suppresses macrophage function, contributing to chemoresistance and poor prognosis in OVCA. In M1 macrophages co-cultured with OVCA cells, deletion of pGSN in OVCA chemoresistant cells (A2780CP and OV90) considerably reduces caspase-3 activation and apoptosis of M1 macrophages, and this response is associated with increased secretion of iNOS and TNFα in M1 macrophages. However, overexpression of pGSN shows opposite effects, suggesting that pGSN inhibits M1 macrophage function [53].

NO induces apoptosis in chemoresistant OVCA cells by increasing reactive oxygen species (ROS) production and decreasing GSH synthesis in response to CDDP. pGSN increases GSH content in OVCA cells by decreasing iNOS production in M1 macrophages; responses that contribute to chemotherapy resistance [53]. Additionally, pGSN also acts as a selective chemotactic factor for M1 macrophages. In chemoresistant conditions where pGSN levels are high, M1 macrophages are selectively attracted to the tumor nest after which they are induced to undergo apoptosis [53]. Thus, the reduction and dysfunction of M1 macrophages by pGSN favors chemotherapy resistance.

### 3.3. pGSN and Other Immune Cells

OVCA-derived exosomes also regulate the function of dendritic cells. EVs in the ascites fluid and plasma of patients with OVCA contain Arginase 1 (ARG1). In a mouse model, ARG1-containing EVs traveled to the draining lymph nodes, taken-up by dendritic cells, and inhibited antigen-specific T cell proliferation [76]. Exosomes containing FasL and tumor necrosis factor-related apoptosis-inducing ligand isolated from OVCA ascites fluid induced apoptosis in dendritic cells [91]. The process of priming CD8+ T cells is based on the presentation of antigens by type 1 dendritic cells (cDC1). cDC1 expresses DNGR-1 that binds to F-actin from dead cell debris. pGSN inhibits DNGR-1 binding to F-actin and reduces the cross-presentation of death cell-associated antigens by cDC1 [54]. These suggest that exosomal pGSN might suppress the function of dendritic cells in OVCA.

Exosomes derived from OVCA cells or malignant ascites reduce NK cell activity by decreasing NKG2D-mediated cytotoxicity [92]. Overexpression of GSN in NK cell line (YTS) inhibits PI3K/Akt signaling, cell growth, colony formation, and invasion as well as promote apoptosis [93]. This implies that pGSN might suppress the NK cell function in OVCA. Taken together. pGSN overexpression in the OVCA tumor microenvironment is detrimental to all anti-tumor immune cells. This suggests pGSN may be a suitable therapeutic target to sensitize tumors to immunotherapy and other alternative treatments.

## 4. Clinical Significance of pGSN

### 4.1. Early Diagnosis

Early diagnosis of OVCA can have a huge impact on the patient’s prognosis [8]. One of the most common tumor markers for OVCA is CA125; however, it has only 50–62% sensitivity in detecting early stage OVCA [94]. Circulating pGSN has been shown to be a reliable and useful marker for cancer diagnosis compared with CA125 [95]. The test accuracy of pGSN is further enhanced when combined with CA125 in a multi-analyte platform. In addition to OVCA, circulating pGSN has been useful in head-and-neck cancer with a detection sensitivity and specificity of 82.7 and 95.6%, respectively. Combining pGSN with sFasL further improved the detection sensitivity up to 90.6% [96].

Patients with colorectal cancer show considerably lower serum GSN levels than those in healthy controls. Receiver operating characteristic (ROC) analysis showed that pGSN (AUC = 0.932) is a more effective diagnostic biomarker for colorectal cancer than CEA (AUC = 0.751) and CA199 (AUC = 0.638), which are currently available for diagnostic use [97]. Blood pGSN levels can be used as a diagnostic tool for early stage OVCA. Patients with stage 1 OVCA had considerably higher preoperative blood pGSN levels than patients with stage > 1 OVCA and healthy controls [95]. With a cutoff value of 81 ng/mL, blood pGSN has a sensitivity of 75% and specificity of 78.4% for OVCA stage 1 detection; a test performance that was better than CA125. In contrast to CA125, pGSN values were unaffected by age [95]. Furthermore, the combination of pGSN and CA125 showed a sensitivity of 100% in the detection of OVCA stage 1, which is superior to ROMA (sensitivity 89%) and CA125 alone [98,99]. These findings suggest that pGSN is a useful marker for detecting OVCA patients at stage 1.

### 4.2. Residual Disease Prediction

Postoperative residual disease and chemotherapy response are important factors affecting OVCA prognosis [100,101]. Imaging has been used to predict whether optimal surgery can be performed in advanced stage OVCA although little success has been achieved [102,103]. Previous studies have also demonstrated the utility of CA125 and HE4 as markers for predicting residual disease although only modest outcomes have been achieved [104,105]. Recently, the presence of residual disease after OVCA surgery has been shown to be reliably predicted by blood pGSN level compared with CA125. pGSN with a cutoff of 64 μg/mL has a sensitivity and specificity of 60%, whereas CA125, with a cutoff of 576.5 U/mL, has a much lower sensitivity and specificity [95]. This suggests that with pre-operative blood analyses of pGSN, clinicians could predict how successful a cytoreductive debulking could be; a strategy that will significantly help improve patient management and overall survival.

### 4.3. Prognostic Marker

Prognostic markers are useful in the clinical management of patients especially cancer patients. Currently, there is no reliable prognostic marker for OVCA patients hence the urgent need to discover novel markers with significant prognostic utility. Previous studies have demonstrated that pGSN is a useful prognostic marker for OVCA, but not for osteosarcoma and prostate, colon, and head and neck cancers [51,106,107,108]. In OVCA, multivariate Cox regression analysis showed that pGSN is a remarkable predictor of progression-free survival (PFS) [95]. Increased pGSN expression in OVCA tumors is considerably associated with shortened PFS and OS. Moreover, patients with low pGSN expression and high CD8+ T cell infiltration had better PFS and OS than patients with high pGSN expression and high CD8+ T cell infiltration. Thus, increased pGSN expression suppressed the prognostic benefits of infiltrated T cells in the OVCA tumor microenvironment [18]. In a study using a Japanese cohort, patients with elevated pGSN levels in OVCA tissue had considerably shorter PFS and OS as well developed chemoresistance. Additionally, pGSN was highly expressed in advanced stages of the tumor compared with the early stages. Just like T cells, increased pGSN expression suppressed the prognostic benefits of infiltrated M1 macrophages [53]. Taken together, pGSN presents as a potential prognostic marker for OVCA that could potentially help revolutionize patient management.

## 5. Summary and Future Research Directions

Immune checkpoint inhibitors (ICIs) have provided remarkable advances in the treatment of lung, breast, and colorectal cancers recently; however, in OVCA treatment, only modest therapeutic success has been achieved. Trials using PD1 (nivolumab and pembrolizumab), PD-L1 (avelumab, aterolizumab, and durvalumab), and CTLA4 (ipilimumab and tremelimumab) antibodies did not demonstrate an improvement in the survival of patients with OVCA [109]. Although ICIs combined with chemotherapy and anti-VEGF antibodies or PARP inhibitors improved survival, high toxicity was observed [109]. Thus, a novel therapeutic approach is needed to sensitize OVCA patients to ICIs and other targeted therapies such as PARP inhibitors. pGSN is highly expressed in chemoresistant OVCA hence immune cells are likely to be suppressed and non-functional. This could explain why ICI has not been successful in OVCA patients given functional immune cells are needed to achieve an effective therapeutic outcome. Anti-pGSN antibodies were shown to be effective in vitro; they attenuated caspase-3 activation and apoptosis, and increased iNOS and TNFα secretion in M1 macrophage treated with conditioned media from chemoresistant OVCA cells [53]. It is also well established that overexpression of HIF1α is associated with OVCA tumor aggressiveness, progression, and metastasis, but inhibition of HIF1α as a therapeutic option has not been successful [110]. Inhibitors of the HIF1α-pGSN binding motif might interfere with their interaction and reduce pGSN production. Thus, a combination treatment of pGSN inhibition and ICI could considerably enhance OVCA patient outcomes. Epigenetic alterations, including DNA methylation and histone modifications, cause chemotherapy resistance in ovarian cancer [111]. DNA methyltransferase 1 and histone deacetylase are involved in epigenetic alteration [112,113]. The histone deacetylase inhibitor trichostatin A increases GSN expression in the breast cancer cell lines MDA231, MCF7, and T47D [114]. Furthermore, lower expression of DNA methyltransferase 1 is associated with increased expression of GSN in gastric cancer cell lines (AGS) [115]. Although pGSN is highly expressed in chemoresistant OVCA tumors compared with chemosensitive ones, the reason for the differential expression is unknown. It’s thus possible that epigenetic changes may be involved in the differential expression of pGSN expression in OVCA, which may contribute to chemoresistance. Whether this is indeed the case and if the dysregulation of DNA methyltransferase and/or Ten-eleven translocation enzymes involved in de-methylates of DNA at CpG islands and cysteine-rich sites, remain to be determined.

There are conflicting reports regarding the expression of gelsolin and the association between gelsolin and prognosis in cancer. Previous studies have shown that GSN expression was decreased in several types of cancers (bladder, breast, lung, and prostate) compared to normal tissues [116,117,118,119]. A previous study has also reported that GSN expression was decreased in OVCA compared to the normal ovarian epithelium [120]. These differences could be due to strategies and methods used in measuring GSN isoforms, sample heterogeneity and the use of normal ovarian tissue as a control sample. Most high-grade serous ovarian carcinomas have been shown to originate from the fallopian tubes [121]. Thus, using normal fallopian tube tissues could present as the appropriate control sample for comparison. In a study by Asare-Werehene et al., 2020, where normal fallopian tube tissues were used as a control, it was observed that pGSN expression was significantly higher in the OVCA tissues [18]. This strengthens the point of using the appropriate control tissues for pGSN analyses. Further, pGSN expression should be examined separately in epithelium and stroma compartments given each compartment could provide a specific prognostic effect. The stroma of colorectal cancer tissues was stained strongly with GSN compared to the epithelium [122]. Similarly, a higher stromal expression of pGSN compared with epithelial expression has been observed in OVCA tissues [18]. High pGSN expression in the stromal area was associated with poor PFS, but not in the epithelial area [18]; leaning toward the argument why tissue compartment specific-analyses is necessary for prognostic impact.

High circulating blood pGSN levels are associated with early-stage ovarian cancer, which has a favorable prognosis [95]. However, increased expression of pGSN in ovarian cancer tissue is associated with late stage, chemotherapy resistance and poor prognosis [18]. Cancer cachexia, characterized by weight loss and muscle wasting, is common in women with advanced ovarian cancer and negatively impacts their survival [123,124]. In a model in which the ovarian cancer cell line ES-2 is injected into the abdominal cavity of female Nod SCID gamma mice, muscle atrophy and generalized grip weakness develop as the ovarian cancer progresses [125]. Circulating blood pGSN levels may be higher in early-stage ovarian cancer because of greater muscle volume compared to advanced ovarian cancer. As inflammation and wasting progresses in OVCA, actin (released as a result of tissue damage) and inflammatory mediators are released into the circulatory system which bind and deplete pGSN. This might contribute to the reason why pGSN is elevated in early stage OVCA but decreased in late stages. It’s thus possible that pGSN-actin complex might be excreted via the urinary system as the tumor advances and urine samples might be a non-invasive form of staging OVCA using pGSN levels. Given pGSN secreted by OVCA cells is carried by exosomes, we hypothesize that circulating exosomal pGSN could be used as a reliable marker of tumor growth, stage and overall survival of the patient. Whether this hypothesis is true remains to be investigated. In conclusion, exosomal pGSN contributes to suppression of immune function and chemotherapy resistance in OVCA. Furthermore, pGSN could be used for the detection of early stage OVCA and prediction of chemoresistance and prognosis. Further research on pGSN will contribute to its usage as a diagnostic marker and potential therapeutic target for OVCA.

## Figures and Tables

**Figure 1 cells-11-03305-f001:**
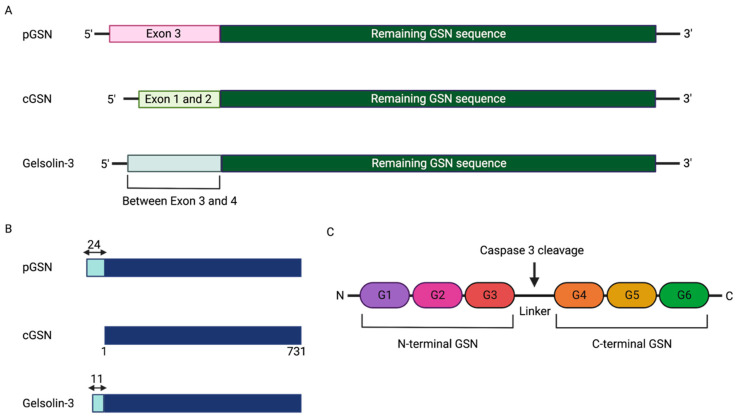
Gelsolin isoforms and structure. (**A**) The pGSN, cGSN, and gelsolin-3 are encoded by a single gene as a result of alternative splicing and different transcriptional initiation sites. These isoforms contain at least 14 exons. Each isoform is characterized by a 5′-end arrangement; A unique 5′ untranslated region of cGSN is formed by exons 1 and 2; The unique 5′-end of pGSN is formed by exon 3. These encode the signal peptide and the first 21 residues of pGSN; The unique 5′-end of Gelsolin-3 is arranged in the area between exon 3 and 4. (**B**) The dark blue color shows the 731 amino acid sequence of cGSN, which is consistent with pGSN and Gelsolin-3. Cyan color shows 24 and 11 different amino acid sequences in pGSN and Gelsolin-3, respectively, compared to cGSN. (**C**) GSN consist of six domains that are named G1-G6. There is a linker between G3 and G4 domain which is cleaved by caspase 3 producing N-terminal GSN and C-terminal GSN. Figure designed using BioRender. pGSN, plasma gelsolin; cGSN, cytoplasmic gelsolin.

**Figure 2 cells-11-03305-f002:**
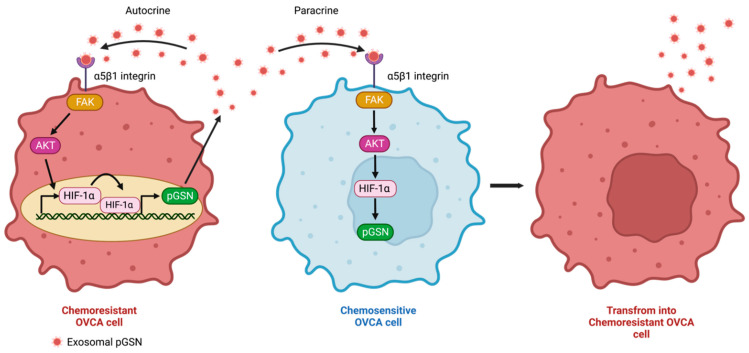
Autocrine and paracrine action of pGSN. Exosomal pGSN derived from chemoresistant OVCA cells transforms chemosensitive cells into chemoresistant OVCA cells via the α5β1 integrins/FAK/Akt/HIF-1α axis (Paracrine action). In chemoresistant OVCA cells, exosomal pGSN increases the promoter region binding of HIF1α and enhances exosomal pGSN production. Thus, exosomal pGSN forms a positive feedback loop of pGSN production via α5β1 integrins/FAK/Akt/HIF-1α axis (Autocrine action). pGSN, plasma gelsolin; FAK, focal adhesion kinase; Akt, Ak strain transforming; HIF-1α, hypoxia-inducible factor 1-alpha. Figure designed using BioRender.

**Figure 3 cells-11-03305-f003:**
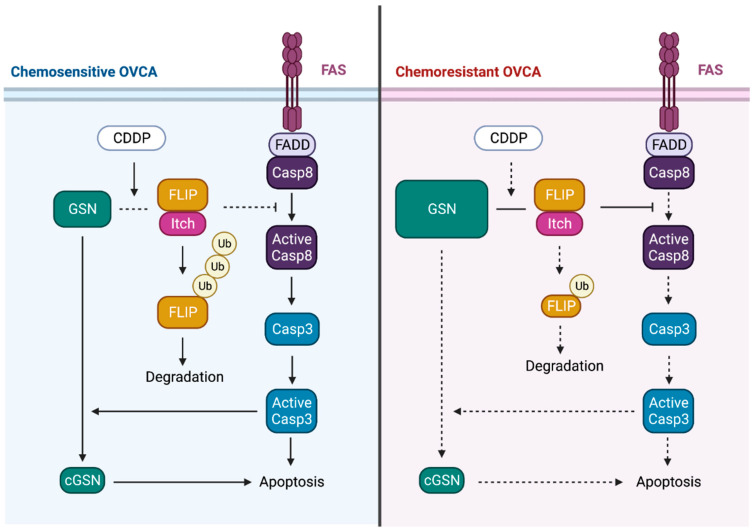
Apoptosis Regulation by pGSN. FLIP and ITCH form a complex with GSN in the chemosensitive OVCA cell. CDDP dissociates GSN from the GSN-FLIP-ITCH complex, leading to FLIP ubiquitination and degradation, caspase-8 and -3 activation, caspase-3-mediated GSN cleavage, and apoptosis. In chemo-resistant OVCA cells, CDDP does not alter the GSN-FLIP-ITCH interaction, attenuating its downstream effects. FLIP, Fas-associated death domain-like interleukin-1b-converting enzyme-like inhibitory protein; ITCH, Itchy E3 ubiquitin protein ligase; Ub, ubiquitin; GSN, gelsolin; CDDP, cis-Diamminedichloroplatinum; FADD, Fas associated via death domain; Casp8, caspase8. (Figure modified from Abedini et al., 2014 [69]. Figure designed using BioRender.

**Figure 4 cells-11-03305-f004:**
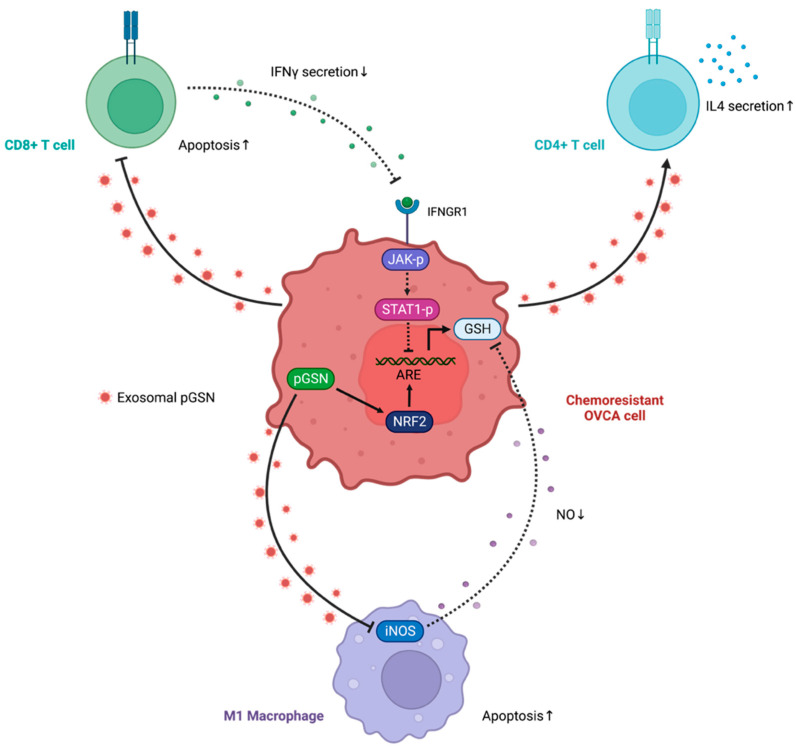
Immune dysfunction induced by pGSN. A chemoresistant ovarian cancer cell-derived exosomal pGSN causes CD8+ T cells to undergo apoptosis via caspase-3 activation. Apoptosis is not induced in CD4+ T cells by pGSN. CD4+ T cells secreted more IL-4, polarized to type 2 helper cells. IFNγ activates the IFNGR1/JAK/STAT1 pathway, decreasing the intracellular GSH levels. pGSN depletes CD8+ T cells and reduces IFNγ secretion. Thus, GSH production in ovarian cancer was increased and contributes to chemoresistance. Exosomal pGSN induces caspase-3 activation and apoptosis of M1 macrophages, which leads to decreased iNOS secretion. Furthermore, pGSN increases GSH content in ovarian cancer cells via decreased iNOS production in M1 macrophages, contributing to chemotherapy resistance. Abbreviations: pGSN, plasma gelsolin; IL-4, interleukin 4; IFNγ, Interferon gamma; IFNGR1, interferon gamma receptor 1; JAK, janus kinase; STAT1, signal transducer and activator of transcription 1; GSH, glutathione; iNOS, inducible nitric oxide synthase. Figure designed using BioRender.

## Data Availability

Not applicable.

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
