# Peer review of "Exosomal Plasma Gelsolin Is an Immunosuppressive Mediator in the Ovarian Tumor Microenvironment and a Determinant of Chemoresistance"

_cells, 2022, doi:10.3390/cells11203305_

Round 1

Reviewer 1 Report

This is a very clearly written review of the relationships between pGSN and OvCa. It is clear to me that it deserves to be published. The authors need to insert the references for their statement at line 137 (Recent studies have demonstrated pGSN overexpression as a key factor in OVCA 137 chemoresistance as well as other malignancies). I believe a summary section of the relationship of pGSN in human clinical OvCa cases would enormously benefit the work.

Author Response

RESPONSE TO REVIEWERS’ COMMENTS

We thank the reviewers for the time they have taken in assessing the acceptability of this manuscript, and for their helpful suggestions. We have taken their concerns into consideration as we revised our manuscript and have provided additional information where needed to address their concerns. We believe that the revised manuscript is a much improved version. Our responses to the specific comments from the reviewers are provided hereafter. 

Reviewer 1

This is a very clearly written review of the relationships between pGSN and OvCa. It is clear to me that it deserves to be published. The authors need to insert the references for their statement at line 137 (Recent studies have demonstrated pGSN overexpression as a key factor in OVCA 137 chemoresistance as well as other malignancies). I believe a summary section of the relationship of pGSN in human clinical OvCa cases would enormously benefit the work.

Response: We agree with the reviewer and have added a reference to line 137 on page 4 as indicated here: Recent studies have demonstrated pGSN overexpression as a key factor in OVCA chemoresistance as well as other malignancies [18,51–54].

Reviewer 2 Report

The review by Onuma et al. deals with exosomal plasma Gelsolin (pGSN) in ovarian cancer and gives an overview about the mechanisms responsible for immunosuppressive effects in the tumor microenvironment and for the induction of chemoresistance in ovarian tumor cells. In addition, the role of pGSN as possible diagnostic and prognostic marker in ovarian cancer patients is discussed; this issue is relevant, since at present a reliable marker is still lacking.

The review includes a quite balanced and critical view of the research area. Importantly, some of the Authors are experts in the field, as demonstrated by several studies published on this topic.

Overall, the review is well-written, and the quality of English language is good. The reference list covers the relevant literature and is up-to-date, including also very recent papers. The four figures are useful in illustrating different issues analyzed in the text and cover the main themes of the review.

I do not find any major omissions, I have only a few minor suggestions that, in my opinion, could improve the quality of the manuscript.

Minor revisions/editing:

- GSN can be expressed in different forms. Although the focus of the review is secreted pGSN and its alteration in ovarian cancer, it may be useful to include some information about the expression of the other GSN isoforms in different tumor types, as well as their possible alteration in pathologic conditions.

- The Authors should also comment more extensively about studies in which GSN was demonstrated to be significantly decreased in different kinds of tumors, including ovarian cancers. This issue should take in consideration the fact that some conflicting reports may actually be explained with the heterogeneity of samples/patients analyzed and of methods utilized to detect different GSN isoforms. Regarding this aspect, it will be necessary to include some additional references.

- Legends to Figures (1A and 2 in particular) are somehow confusing: they could be improved and made clearer.

- Page 5, first line of paragraph 2.2: the sentence is incomplete.

- The full name for CDDP should be indicated not only in Legend to Figure 3 but also the first time CDDP appears in the text (page 6).

- Page 8, second line: MI macrophages should be M1 macrophages, I guess.

Author Response

RESPONSE TO REVIEWERS’ COMMENTS

We thank the reviewers for the time they have taken in assessing the acceptability of this manuscript, and for their helpful suggestions. We have taken their concerns into consideration as we revised our manuscript and have provided additional information where needed to address their concerns. We believe that the revised manuscript is a much improved version. Our responses to the specific comments from the reviewers are provided hereafter. 

Reviewer 2:

The review by Onuma et al. deals with exosomal plasma Gelsolin (pGSN) in ovarian cancer and gives an overview about the mechanisms responsible for immunosuppressive effects in the tumor microenvironment and for the induction of chemoresistance in ovarian tumor cells. In addition, the role of pGSN as possible diagnostic and prognostic marker in ovarian cancer patients is discussed; this issue is relevant, since at present a reliable marker is still lacking.

The review includes a quite balanced and critical view of the research area. Importantly, some of the Authors are experts in the field, as demonstrated by several studies published on this topic.

Overall, the review is well-written, and the quality of English language is good. The reference list covers the relevant literature and is up-to-date, including also very recent papers. The four figures are useful in illustrating different issues analyzed in the text and cover the main themes of the review.

I do not find any major omissions, I have only a few minor suggestions that, in my opinion, could improve the quality of the manuscript.

Minor revisions/editing:

- GSN can be expressed in different forms. Although the focus of the review is secreted pGSN and its alteration in ovarian cancer, it may be useful to include some information about the expression of the other GSN isoforms in different tumor types, as well as their possible alteration in pathologic conditions.

Response; We acknowledge the suggestions of the reviewer and have included information about the expression of other GSN isoforms in other tissues. Also, it is very difficult detecting only cGSN expression in tumor tissues given the antibodies and strategies used in those studies target the c-terminal region which is similar in both pGSN and cGSN. Thus, could be classified as total GSN rather than cGSN. Recent studies have specifically targeted pGSN since the antibodies used target the extra 24 amino acid sequence on the N-terminal of GSN which is absent in the cGSN. Given these challenges, it’s difficult including exclusive information on the expression of cGSN in different tumor types, as well as their possible alteration in pathologic conditions. We have thus explained this on page 3, as below.

Page 3 line 262

“Detection of only cGSN expression in tumor tissues is challenging given the antibodies and strategies used target the c-terminal regions which is similar in both pGSN and cGSN. Thus, could be classified as total GSN rather than cGSN. Recent studies have specifically targeted pGSN with antibodies that binds to epitopes at the extra 24 amino acid sequence on the N-terminal of GSN which is absent in the cGSN [18,53].”

- The Authors should also comment more extensively about studies in which GSN was demonstrated to be significantly decreased in different kinds of tumors, including ovarian cancers. This issue should take in consideration the fact that some conflicting reports may actually be explained with the heterogeneity of samples/patients analyzed and of methods utilized to detect different GSN isoforms. Regarding this aspect, it will be necessary to include some additional references.

Response; We appreciate the reviewer’s suggestions. We believe that the conflicting report is due to the heterogeneity of samples/patients analyzed and of methods utilized to detect different GSN isoforms. We have added the following in “Summary and Future research directions”.

Page 11 line732

There are conflicting reports regarding the expression of gelsolin and the association between gelsolin and prognosis in cancer. Previous studies have shown that GSN expression was decreased in several types of cancers (bladder, breast, lung, and prostate) compared to normal tissues [116–119]. A previous study has also reported that GSN expression was decreased in OVCA compared to the normal ovarian epithelium [120]. These differences could be due to strategies and methods used in measuring GSN isoforms, sample heterogeneity and the use of normal ovarian tissue as a control sample. Most high-grade serous ovarian carcinomas have been shown to originate from the fallopian tubes [121]. Thus, using normal fallopian tube tissues could present as the appropriate control sample for comparison. In a study by Asare-Werehene et. al., 2020, where normal fallopian tube tissues were used as a control, it was observed that pGSN expression was significantly higher in the OVCA tissues [18].  This strengthens the point of using the appropriate control tissues for pGSN analyses. Further, pGSN expression should be examined separately in epithelium and stroma compartments given each compartment could provide a specific prognostic effect. The stroma of colorectal cancer tissues was stained strongly with GSN compared to the epithelium [122]. Similarly, a higher stromal expression of pGSN compared with epithelial expression has been observed in OVCA tissues [18]. High pGSN expression in the stromal area was associated with poor PFS, but not in the epithelial area [18]; leaning toward the argument why tissue compartment-specific analyses is necessary for prognostic impact.

- Legends to Figures (1A and 2 in particular) are somehow confusing: they could be improved and made clearer.

Response; We have changed the descriptions of Figures 1A and 2 to make them clearer as below.

Figure 1A line 281

The pGSN, cGSN, and gelsolin-3 are encoded by a single gene as a result of alternative splicing and different transcriptional initiation sites. These isoforms contain at least 14 exons. Each isoform is characterized by a 5'-end arrangement; A unique 5' untranslated region of cGSN is formed by exons 1 and 2; The unique 5’-end of pGSN is formed by exon 3. These encode the signal peptide and the first 21 residues of pGSN; The unique 5’-end of Gelsolin-3 is arranged in the area between exon 3 and 4.

Figure 2 line 358

Exosomal pGSN derived from chemoresistant OVCA cells transforms chemosensitive cells to chemoresistant cells via the α5β1 integrins/FAK/Akt/HIF-1α axis (Paracrine action). In chemoresistant OVCA cells, exosomal pGSN increases the promoter region binding of HIF1α and enhances exosomal pGSN production. Thus, exosomal pGSN forms a positive feedback loop of pGSN production via α5β1 integrins/FAK/Akt/HIF-1α axis (Autocrine action). 

- Page 5, first line of paragraph 2.2: the sentence is incomplete.

Response; We have made the following corrections.

Page 5 line 367

Chemotherapeutic agent-induced apoptosis is an important indicator of chemoresponsiveness of cancer cells.

- The full name for CDDP should be indicated not only in Legend to Figure 3 but also the first time CDDP appears in the text (page 6).

Response; We have described the name of the CDDP as below.

Figure 3 line 434

Apoptosis Regulation by pGSN. FLIP and ITCH form a complex with GSN in the chemosensitive OVCA cell. CDDP dissociates GSN from the GSN-FLIP-ITCH complex, leading to FLIP ubiquitination and degradation, caspase-8 and -3 activation, caspase-3-mediated GSN cleavage, and apoptosis. In chemo-resistant OVCA cells, CDDP does not alter the GSN-FLIP-ITCH interaction, attenuating its downstream effects. FLIP, Fas-associated death domain-like interleukin-1ß-converting enzyme-like inhibitory protein; ITCH, Itchy E3 ubiquitin protein ligase; Ub, ubiquitin; GSN, gelsolin; CDDP, cis-Diamminedichloroplatinum; FADD, Fas associated via death domain; Casp8, caspase8. (Figure modified from Abedini et al, 2014 [figure 6; 19]. Figure designed using BioRen-der.

Page 6 line 419

FLIP interacts with an E3 ligase (i.e., ITCH) in response to cis-Diamminedichloroplatinum (CDDP), resulting in ubiquitination and proteasomal degradation. This leads to apoptosis through the activation of caspase-8 and caspase-3 [65,66].

- Page 8, second line: MI macrophages should be M1 macrophages, I guess.

Response; We have made the following corrections.

Page 8 line 557

Exosomes can cause M1 macrophages dysfunction and promote M1 to M2 polarization in OVCA [88–90].